# Message in a Bottle: Endothelial Cell Regulation by Extracellular Vesicles

**DOI:** 10.3390/cancers14081969

**Published:** 2022-04-13

**Authors:** Claudia Palazzo, Alessio D’Alessio, Luca Tamagnone

**Affiliations:** 1Sezione di Istologia ed Embriologia, Dipartimento di Scienze della Vita e Sanità Pubblica, Università Cattolica del Sacro Cuore, 00168 Roma, Italy; claudia.palazzo@unicatt.it (C.P.); alessio.dalessio@unicatt.it (A.D.); 2Fondazione Policlinico Universitario “Agostino Gemelli”, IRCCS, 00168 Roma, Italy

**Keywords:** endothelial cells, exosomes, extracellular vesicles, miRNA, angiogenesis, cancer, atherosclerosis, diabetes

## Abstract

**Simple Summary:**

Elucidating the role of extracellular vesicles (EVs) in the communication mechanisms between cancer and endothelial cells (ECs) within the tumor microenvironment is an exciting challenge. At the same time, due to their ability to convey bioactive molecules, EVs may be potentially relevant from a therapeutic perspective for diverse vascular pathologies.

**Abstract:**

Intercellular communication is a key biological mechanism that is fundamental to maintain tissue homeostasis. Extracellular vesicles (EVs) have emerged as critical regulators of cell–cell communication in both physiological and pathological conditions, due to their ability to shuttle a variety of cell constituents, such as DNA, RNA, lipids, active metabolites, cytosolic, and cell surface proteins. In particular, endothelial cells (ECs) are prominently regulated by EVs released by neighboring cell types. The discovery that cancer cell-derived EVs can control the functions of ECs has prompted the investigation of their roles in tumor angiogenesis and cancer progression. In particular, here, we discuss evidence that supports the roles of exosomes in EC regulation within the tumor microenvironment and in vascular dysfunction leading to atherosclerosis. Moreover, we survey the molecular mechanisms and exosomal cargoes that have been implicated in explanations of these regulatory effects.

## 1. Introduction

“The music that the nucleus hears” is how Pierre McCrea portrays cell–cell communication [1]. Indeed, the cellular transcriptomic profile, in any tissue context, is largely the product of extracellular signals dispatched by neighboring cells. Thereby, various mechanisms of cell–cell communication are key regulators of cell behavior and metabolism, beginning at the stage of embryonic development to adult tissue homeostasis, and their defects are accountable for human diseases [2].

It is pivotal to reckon the multi-factorial complexity of cell–cell signaling modes, including factors that act at a short distance (paracrine and synaptic signaling), as opposed to long-range endocrine signaling. In addition, secreted ligands may self-regulate signal-emanating cells, known as autocrine regulation [3], or deploy intracrine activity independent of their extracellular release [4]. Cells can also communicate by contact, as in the case of juxtacrine signaling between membrane-bound ligands and cognate receptors on the surface of adjacent cells. Notably, transmembrane molecules can exert dual functions, acting both as ligands and as receptors, with bi-directional signals exchanged between neighboring cells, and therefore greatly increasing the plasticity of intercellular communications. In particular, the term “reverse signaling” refers to the setting in which a known membrane-bound ligand also functions as a receptor, triggering intracellular pathways in the expressing cell. Considerable evidence supports this kind of signaling mechanism, especially mediated by semaphorin [5] and ephrin [6] family members.

More recently, extracellular vesicles (EVs) have emerged as novel major players in cell–cell communication. They are released by almost all cell types, including endothelial and epithelial cells, blood cells, tumor cells, dendritic cells, and neurons, in both physiological and pathological conditions [7]. EVs have been isolated from diverse body fluids, for example, blood, saliva, semen, seminal plasma, breast milk, synovial fluid, nasal secretion, urine, amnion, ascites, and cerebrospinal fluids [8].

EV secretion plays an important role in cellular communication, thereby influencing the behavior of the cells with which they interact, by conveying material and information from donor cells. Here, we review the most recent evidence about communication pathways mediated by exosomes between endothelial cells and the surrounding cell populations, with a special focus on the tumor microenvironment, and on pathological conditions such as diabetes and atherosclerosis.

## 2. Exosomes and Microvesicles

Based on the current knowledge about their biogenesis, EVs can be subdivided into two main categories, i.e., exosomes and microvesicles, with the latter also including apoptotic bodies and oncosomes that are the largest known extracellular vesicles [9]. The term exosome was initially applied to vesicles of unknown origin released from a variety of cultured cells [10]. Subsequently, this term has been adopted to define a population of membrane vesicles with a typical cup-like shape [11], with size ranging between 30 and 150 nm in diameter, which are released by reticulocytes during differentiation [12]. It is worthwhile to stress that cells release distinct subpopulations of exosomes with heterogeneous sizes and compositions [13]. Instead, microvesicles (MVs), formerly called ”platelet dust”, were first described as subcellular material originating from platelets in normal plasma and serum [14]. In contrast to exosomes, MVs represent a more heterogeneous population with sizes ranging from 100 to 1000 nm in diameter, and up to 10 μm in the case of oncosomes [12].

The aforementioned EVs have different modes of biogenesis. Exosomes are intraluminal vesicles formed by the inward budding of endosomal membrane during maturation of multivesicular bodies (MVBs) [12]. Intraluminal vesicle formation constitutes the starting point of exosome biogenesis, and mostly requires the endosomal sorting complexes required for transport (ESCRT) machinery [15]. Then, intraluminal vesicles (and thereby exosomes) are released into the extracellular environment upon MVB fusion with the cell surface, a mechanism that was first described in the 1980s [16,17] and has been speculated to depend on SNARE proteins and synaptotagmin family members [18]. It is known that MVBs are primarily destined to fuse with lysosomes for degradation; but in fact, this fate can be skipped by mechanisms that control the balance between the degradative and secretory capacity of MVBs. Although the molecular details have remained largely unknown, the first level of regulation is thought to be exerted on the sorting machinery of MVBs. For instance, while different components of the ESCRT complex are commonly associated with the degradation of MVBs, the syndecan–syntenin–ALIX pathway seems to be restricted to exosome secretion [13,19]. In addition, the ESCRT-independent mechanisms for sorting exosomes from MVBs implicate tetraspanin microdomains and ceramide-enriched lipid rafts [20,21].

As opposed to exosomes, MVs are usually generated by the outward budding and fission of the plasma membrane, and the subsequent release of vesicles into the extracellular space [22]. These processes seem to occur selectively in the lipid-rich microdomains of the cell membrane, such as in lipid rafts or caveolae [23]. Moreover, an alternative mechanism of MV production implicates the ESCRT complex, which is mostly associated with endosome and exosomes biogenesis [24]. The varied mechanisms leading to the secretion of diverse EVs, as suggested by Guillaume van Niel [12], impose a different timing between the generation of exosomes and MVs. The release of MVs is probably faster because their release would directly follow generation and fission. By contrast, the release of exosomes requires additional steps to sort cargoes to MVBs, and then to ILVs, and extra steps to target MVBs to the plasma membrane and finally deploy their secretion [12].

Importantly, EV cargoes, which include proteins, lipids, and nucleic acids, broadly reflect the nature and the status of the cells of origin and are, thereby, considered to be “cell biopsies” [8,25]. In recent years, numerous have studies have focused on providing an exhaustive and comprehensive characterization of the content of EVs, but the scenario remains largely unexplored. The lipids generally enriched in EVs are sphingomyelin, cholesterol, phosphatidylserine, ceramide, and glycosphingolipids, which confer a bilayer structure similar to that of membrane raft microdomains. Proteins commonly found in EVs include molecules responsible for vesicle biogenesis and trafficking, such as tetraspanins (CD81, CD9, and CD63), stress-response proteins (heat shock proteins and HSP90), members of the ESCRT complex (Tsg101 ad Alix), and proteins involved in membrane fusion (Rabs and ARF6) [25,26]. For years, these proteins were considered to be a prerogative of exosomes; however, it is now generally agreed that they can also be found in larger vesicles, such as MVs [25]. In addition, several reports have challenged the idea of a uniform representation of these proteins in EVs across different conditions and cells of origin [15]. Importantly, EVs also carry transmembrane signaling proteins (exposed on the surface with similar topology as in producer cells), and membrane-tethered secreted signals, such as cytokines and growth factors [27]. Much attention has focused on the ability of EVs to incorporate and vehicle genetic material, such as small and long coding and noncoding RNA (mRNA, miRNA, circRNA, and lncRNA) [28,29,30]. Certain studies have also reported about genomic and mitochondrial DNA found in EVs [31], although mechanisms of DNA packaging into these vesicles are still unclear. Interestingly, the lipid bilayer of EVs encapsulates these genetic cargoes, protecting them from enzymatic digestion [32]. Thus, EVs represent a new mechanism of genetic exchange between cells. Moreover, EVs purified from “liquid” biopsies of patients’ biological fluids have provided biomarkers of clinical relevance [33]. In particular, circulating EVs purified from cancer patients offer a comprehensive representation of intra-tumor heterogeneity (including cells found in metastatic sites). At the same time, the detection of EVs released by different cell types calls for methods that can sort tumor-derived EV subpopulations on the base of specific markers [34].

## 3. The Role of EVs in Cell–Cell Communication

The important roles of EVs in cell communication have been extensively documented. As mentioned above, EVs can convey different types of biological information to recipient cells, as they carry a spectrum of constituents of the cell of origin, including DNA, RNA, lipids, metabolites, cytosolic, and cell surface proteins. For EVs to act as signaling vehicles that elicit functional responses and promote phenotypic changes, it is imperative that they reach the recipient cells and deliver their contents; however, this process is complex, and the underlying mechanisms remain largely unknown (Figure 1).

In principle, EVs elicit functional responses simply by engaging and activating signaling receptors expressed on the surface of recipient cells, regardless of their intracellular uptake [35]. EVs may be internalized by clathrin-mediated or clathrin-independent endocytosis, such as macropinocytosis and phagocytosis [36,37], as well as through endocytosis via caveolae and lipid rafts [12,35,38]. Although EVs generally reflect the content of the releaser cell, proteomic studies have suggested that specific protein-sorting mechanisms were associated with exosome biogenesis and cargo loading. Moreover, EV cargo heterogeneity can reflect the organ and tissue of origin, or their release from cancer cells [39], giving EVs distinctive properties such as tropism to certain organs and favored uptake by specific cell types. Cancer-derived exosomes often carry genetic variants, which may also regulate their ontogeny. For example, it has been shown that oncogenic epidermal growth factor receptor (EGFR) and EGFR variant III (EGFRvIII) were detectable in EVs isolated from tumor cells both in vitro and in vivo. Moreover, the expression of constitutively active EGFRvIII in glioblastoma (GBM) cells has been shown to have an impact on the spectrum of EV protein cargoes, leading to the enrichment of proteins putatively supporting cancer invasion [40]. Surprisingly, upon treatment with inhibitors of EGFR kinase, EVs were enriched in phosphorylated EGFR (in sharp contrast with the cytosolic fraction), as well as in their content of genomic DNA. These findings support the analysis of EV cargoes as potential biomarkers of the efficacy of targeted therapy [41]. Intriguingly, exosomes derived from breast and prostate cancer cells have been found to instigate the acquisition of neoplastic behavior in non-tumoral cells, through the transfer of miRNA cargoes [42,43]. Other studies support the idea that cancer cell plasticity may be attributed, in part, to EVs, for example, exosomal miR-200 from metastatic breast cancer cells fosters epithelial-to-mesenchymal transition (EMT), tumor aggressiveness, and metastasis [44]. Moreover, miR-105 from breast cancer cell-derived exosomes suppresses endothelial tight junction zonular occludens-1 expression, promoting metastasis by impairing the integrity of blood vessels and enhancing vascular permeability [45]. Moreover, mRNAs transferred by exosomes have been implicated in angiogenesis and extracellular matrix remodeling in the tumor microenvironment. For example, matrix metalloproteinase (MMP) 1 mRNA of ascites-derived ovarian cancer (OC) patients affected mesothelial barrier integrity and promoted peritoneal metastatic dissemination [46]. Finally, proteins exposed on the surfaces of EVs have been shown to trigger signaling cascades through receptor/ligand interactions, independent of vesicle internalization [35,47]. For example, transforming growth factor-β (TGFβ), expressed on the surface of cancer cell-derived exosomes, induced fibroblast activation [48]. Moreover, exosomes were found to carry semaphorins, a family of soluble and membrane-bound proteins identified as potent chemorepulsive axon guidance cues during development, playing a key role in neural network formation. Interestingly, SEMA7A, a member of the semaphorin family of guidance cues [49], was found on the surface of GBM stem cell-derived exosomes; notably, by interacting with integrin β1 receptor, this signal activated focal adhesion kinase into glioblastoma stem cells, enhancing their motility and tumor aggressiveness [50].

## 4. Regulation of Endothelial Cells Functions by Tumor-Derived Exosomes

As mentioned above, EVs can either promote or inhibit new blood vessel formation, depending on their cargo and the types of upstream stimuli acting on the releaser cell. In the last decade, many studies have documented the role of EVs in angiogenesis and emphasized their therapeutic potential [25]. Angiogenesis refers to the process by which new blood vessels sprout from a pre-existing vascular network, and occurs throughout life in both health and disease [51]. In healthy tissues, angiogenesis is tightly regulated by a precise balance between stimulatory and inhibitory signals [52]. From a mechanistic viewpoint, abundant pro- and anti-angiogenic factors, extracellular matrix components, and intracellular signaling cascades control this process. In particular, the EC-specific mitogen vascular endothelial growth factor (VEGF) is a major inducer of vascular growth during development and tissue repair as well as a key regulator of physiological and pathological angiogenesis [53]. Notably, angiogenesis is frequently hijacked to support tumor growth and metastatic progression [54]. In this regard, a key role in the formation of aberrant vessels is played by an imbalance between pro- and anti-angiogenic factors, particularly seen in hypoxic tissues [55], in tumors, and in other pathological contexts, such as atherosclerosis, corneal neovascularization, rheumatoid arthritis, or ischemic diseases [23,56]. As previously observed by Judah Folkman, neovascularization is necessary to allow the expansion of a primary tumor mass and metastasis [57,58]. In addition, cancer cells have the singular ability to form vascular-like structures that can support the nutritional needs of a tumor independently of neoangiogenesis or ECs, a phenomenon known as vasculogenic mimicry [56]. Indeed, cancer cell behavior largely depends on signals from the microenvironment, as well as on the continuous supply of oxygen and nutrients [59]. In fact, while, initially, blood vessels in the tumor microenvironment are scarce, ECs are induced to exit their quiescent condition in response to cancer-derived cues, a mechanism known as “angiogenic switch”, which enables vessel sprouting to form a new capillary network [60]. Exosomes have been gaining increasing importance in this regard, as potential systems regulating cell–cell communication within the tumor microenvironment. In fact, though most types of cells release exosomes, tumor cells are a particularly active source of these EVs, especially in hypoxic conditions [61]. Interestingly, it has been shown that the plasma of cancer patients, particularly in the presence of metastases, carry higher amounts of exosomes as compared with that of healthy donors [8,62], suggesting that EVs retrieved from biological fluids (liquid biopsies) may have a prospective application in cancer management [63].

Tumor-derived exosomes (TEXs) might be decisive to understand the mechanisms regulating tumor angiogenesis, as suggested by studies showing their capacity to modulate ECs’ phenotype, proliferation, migration, and tubulogenesis, both in vitro and in vivo [64,65,66,67,68] (Figure 2). Moreover, Mao et al. [64] intriguingly reported that exosomes derived from esophageal squamous cell carcinoma cultured under hypoxic conditions were potent stimulators of ECs proliferation, migration, invasion, tube formation, and significantly enhanced tumor growth and lung metastasis in nude mice tumor models, with respect to exosomes harvested under normoxic conditions [64].

Although our understanding of the molecular mechanisms underlying angiogenesis regulation by exosomes is still limited, “omic” studies have highlighted some of the pivotal protein and RNA mediators of this activity [69,70]. The angiogenic potential of tumor-derived exosomes towards ECs has been associated with exosome-carried pro-angiogenic proteins (Table 1). For example, VEGF, TGFbeta, bFGF, IL-6, IL-8, as well as tissue inhibitor matrix metalloproteinase (TIMP)-1 and -2 have been found to be enriched in exosomes derived from GBM cells, and were reported to affect angiogenesis and to increase tumor malignancy [66,71]. Angiogenesis is also promoted by TEXs carrying matrix metalloproteinases exposed on their surface, especially MMP-2, MMP-9, and MMP-13, which can reshape the extracellular matrix, promoting angiogenesis and metastatic dissemination [66,71,72,73]. In particular, MMP-13, which is abundant in nasopharyngeal carcinoma exosomes, actively promotes proliferation and tube formation in human umbilical vein endothelial cells (HUVECs) [73]. Semaphorin 3A (SEMA3A), a known EC-regulatory factor [74], was found on the surface of GBM-derived EVs, and associated with enhanced vessel permeability in the brain [27]. Furthermore, exosomes derived from head and neck squamous cell carcinoma cells, enriched in the proangiogenic urokinase-type plasminogen activator (uPA), coagulation factor III, and MMP-9, promoted HUVEC proliferation in vitro and the formation of vascular structures in vivo [75].

It has also been found that tumor-derived exosomes borne from ascites of colorectal carcinoma patients were enriched in tetraspanin-8 and plexin B2, which have been implicated in angiogenesis [76]. Interestingly, it has been shown that TEX release by lung adenocarcinoma cells depends on the transmembrane protein sortilin; this mechanism mediates the transfer of EGFR into ECs, resulting in a subsequent upregulation of angiogenic proteins [77]. Biagioni et al. [78] further showed that exosomes released from both A375 and M6 melanoma cells induced the upregulation of VE-cadherin, uPAR, and EGFR levels in both mature ECs and endothelial progenitor cells, along with an increase in ERK1,2 phosphorylation [78]. It was thereby supposed that EGFR expression in ECs of tumor vessels [79] could derive from exosomes released by malignant cells. Annexin II carried by TEXs was found by Maji et al. [80] to act as an angiogenesis-promoting protein in breast cancer in a tPA-dependent manner, although the underlying mechanisms have been not fully elucidated [80]. Interestingly, it has been reported that the enzyme heparanase is a strong promoter of TEX release; furthermore, heparanase impacts on exosome protein cargo, fostering higher levels of angiogenic factors (syndecan-1, VEGF, and hepatocyte growth factor), as well as increased induction of EC tube formation [81].

As indicated by several studies, mRNAs, miRNAs, and other non-coding RNAs transferred by TEXs are responsible for reprogramming recipient cells, including ECs [82] focused on in this article (Table 2). For example, miR-25-3p, which has been associated with poor prognosis and metastatic dissemination in colorectal cancer (CRC) patients [82], can be transferred to ECs by means of exosomes, and can contribute to the disruption of the endothelial barriers and angiogenesis. Exosomal miR-25-3p acts by regulating the expression of VEGFR2, ZO-1, occluding, and claudin-5 in ECs, through targeting Krüppel-like factor 2 (KLF2) and KLF4, and consequently promoting vascular permeability. Moreover, it enhanced liver and lung metastasis in CRC murine models [83]. In addition, miR-25-3p contained in CRC-secreted exosomes has been reported to induce the formation of pre-metastatic niches at distant sites, by promoting angiogenesis and disrupting tight junctions of vein ECs.

Moreover, exosomal miR-105 and miR-181c, released from breast cancer cells, can disrupt endothelial and blood–brain barriers during the early pre-metastatic stage, resulting in increased vascular permeability and metastasis formation [84]. Actually, the identification of biomarkers involved in pre-metastatic niche formation is of potential value for diagnosis, prognosis, and prevention of metastasis [37]; exosomes are considered to be a key cancer-derived structure priming pre-metastatic niche formation in distant organs [85]. Additionally, miR-130a has been found to be significantly upregulated in gastric cancer (GC) and in the derived exosomes [86]; notably, exosome-borne miR-130a promoted angiogenesis and tumor growth by targeting c-MYB, both in vivo and in vitro, supporting its relevance as a potential biomarker for monitoring GC progression [87]. In addition, miR-155 and miR-135b, found in TEXs derived from GC, have been positively implicated in angiogenesis. In particular, exosome-borne miR-155 enhances new vessels formation in vitro through inhibition of Forkhead box O3 (FOXO3a) expression, which is also known to sustain tumor progression. Exosomal miR-135b exerts the same function by suppressing FOXO1 levels [88,89]. Recent data unveiled that OC cells released, into the microenvironment, EVs that contained miR-205, which has been previously found to promote invasion and metastasis in many human cancers [90]. The authors found that exosome-borne miR-205 also acted in a paracrine manner to promote angiogenesis and tumor growth in a mouse model. Moreover, miR-205 was enriched in the ECs of tumor vessels, and its levels correlated positively with microvessel density in OC samples. It has also been shown that miR-205 induces angiogenesis by regulating the PTEN-AKT pathway [91]. Another study revealed the communication between hypoxic papillary thyroid cancer cells and ECs. It was shown that miR-21-5p packaged in exosomes released by hypoxic papillary thyroid cancer cells directly targeted and suppressed TGFBI and COL4A1 expression in ECs, thereby increasing endothelial tube formation and angiogenesis in vitro and in vivo [92].

Beyond miRNAs, small circular RNAs (circRNAs) have been implicated in multiple cancer-related biological processes, including cell growth [93], metastasis [94], and apoptosis [95]. Likewise, miRNAs and circRNAs are carried by exosomes and can be detected in patients’ blood and urine samples, suggesting that they may represent additional noninvasive markers for human cancer diagnosis [96]. Recently, the exosome-mediated transfer of circRNAs was highlighted as a novel mechanism of cancer progression [97]. In particular, Huang and colleagues found that the exosomal circRNA-100338 was expressed in highly metastatic as compared with low-metastatic hepatocellular carcinoma cells (HCCs). They showed that circRNA-100338 induced HUVEC proliferation, vessel formation in vitro, and increased permeability. Moreover, exosomal circRNA-100338 enhanced the metastatic ability of HCC cells in vivo [98]. It was also reported that circ-IARS expression was upregulated in pancreatic cancer tissues and in circulating exosomes in patients with a metastatic disease. Li et al. [99] found that circ-IARS entered HUVECs through exosomes, and thereby promoted tumor invasion and metastasis. In particular, circ-IARS expression has been positively correlated with vascular invasion, as well as lymph node and liver metastasis; at the same time, it was inversely associated with patient survival after surgery. Indeed circ-IARS induced significant downregulation of miR-122 and ZO-1 expression, while it upregulated RhoA and RhoA-GTP levels, increased F-actin expression, focal adhesion formation, and enhanced endothelial permeability; thus, promoting tumor invasion and metastasis [99]. The wide spectrum of EC-regulating small RNA species found in the exosomes released by different tumor cells might reflect tissue-specificity. However, this research field needs further development, and future studies should address the consistency of these findings across human cancers, potentially identifying common biomarkers and the most significant miRNAs regulating tumor vasculature.

**Table 2 cancers-14-01969-t002:** Exosomes-borne small RNAs and their functional roles in EC dysregulation.

Cellular Origin of Exosomes	Small-RNA	Functions	REFs
Colorectal carcinoma	miR-25-3p	Disrupts endothelial barrier ↑ Angiogenesis ↑ Metastasis dissemination Induces the formation of pre-metastatic niches	[82,83]
Breast cancer	miR-105 and 181c	Promote vascular permeability and metastasis	[84]
Gastric cancer	miR-130a miR-155 miR-135b	Promote angiogenesis and tumour growth ↑ Generation of new vessels in vitro Inhibit FOXO3a ↑ Growth of blood vessels ↓ FOXO1	[86,87,88,89]
Ovarian cancer	miR-205	Induces angiogenesis via PTEN-AKT ↑ Metastasis	[91]
Papillary thyroid cancer	miR-21-5p	↓ TGFBI and COL4A1 ↑ Endothelial tube formation	[92]
Hepatocellular carcinoma	circRNA-100,338	↑ Metastatic ability ↑ Angiogenesis ↑ Cell proliferation ↑ Permeability and vascular mimicry	[98]
Pancreatic cancer	circ-IARS	Promote tumour invasion and metastasis	[99]

Symbols Legend: ↑ increase; ↓ decrease.

Intriguingly, multiple studies support the idea that tumor-derived exosomes can “educate” additional neighboring cells beyond ECs, such as mesenchymal stem cells [100], monocytes [101], and dendritic cells [102]; notably, these TME components are well known to have a role in angiogenesis regulation [103]. Moreover, in a mouse xenograft model, it was found that exosomes released by adipocytes in HCC promoted, in turn, tumor growth and angiogenesis [104]. Experiments in HUVECs indicated that these exosomes upregulated the expression of pro-angiogenic molecules ANG1 and FLK1/VEGFR2, while downregulating anti-angiogenic VASH1 and TSP1. In addition, tube formation in vitro was significantly increased in the presence of exosomes found in adipocyte-conditioned medium [104]. However, the underlying molecular mechanisms have not been addressed.

In addition to responding to exosome-borne signals, ECs release exosomes themselves, which can mediate the communication with other cells, and can act in an autocrine manner, to modify the microenvironment. For instance, van Balkom and colleagues employed an endothelial cell line releasing protein- and RNA-containing exosomes to investigate the activity of EC exosomes [105]. They found that miR-214 was enriched in these EVs, especially in response to cellular stress such as hypoxia or inflammatory cytokines, and that it played a crucial role in paracrine signaling between ECs. In fact, EC-derived exosomes stimulated migration and angiogenesis of recipient endothelial cells, whereas exosomes derived from miR-214-depleted ECs failed to stimulate these processes [105]. Moreover, it has been found that Yes-associated protein 1 (YAP1), which is a major regulator of cancer cell proliferation, was also implicated in sustaining EC growth and tube formation, and controlled EC exosomes release [106]. In fact, YAP1 depletion (or functional inhibition) in ECs led to a rebound increase in released exosomes carrying the long non-coding RNA (lncRNA) MALAT1. Notably, a direct exosomal-mediated transfer of MALAT1 to hepatocarcinoma cells induced matrix invasion via ERK1/2 signaling. These findings underscore a potential key role of EC exosomes accounting for the increased invasiveness observed in response to therapies targeting the tumor vasculature [106].

Finally, a key role during metastatic dissemination is mediated by the lymphatic system [107,108], which is a mechanism also favored by the higher permeability of lymphatic vessels as compared with blood vessels. Several secreted factors released by lymphatic ECs (LECs) have been suggested to regulate cancer cells and LEC crosstalk [109,110]. It has been reported that the transcriptional regulator ELK3 found in LECs promoted the expression of pro-oncogenic miRNAs and suppressed anti-oncogenic miRNAs, thereby controlling the signaling cargo transferred to tumor cells through exosomes. In fact, LEC-derived exosomes significantly increased the migration and invasion of MDA-MB-231 cells in vitro, and this was dependent on ELK3 expression in LECs, featuring a major mechanism of communication between the TME and cancer cells promoting metastasis [111]. Moreover, suppression of ELK3 in LECs diminished the ability of LECs to promote tumor growth and metastasis, in vivo.

## 5. Endothelial Regulation by Exosomes in Atherosclerosis

Due to their pivotal role in the regulation of vascular homeostasis, EV-mediated functions have been implicated in other major endothelial dysfunctions [112]. In particular, here, we focus on the pathogenesis of atherosclerotic cardiovascular disease [113,114]. As reported above for tumor angiogenesis, most studies have aimed at the identification of exosome-mediated regulatory mechanisms and potential therapeutic targets. Moreover, circulating exosomes may carry biomarkers valuable for monitoring disease progression at a systemic level.

Briefly, atherosclerosis (AS) is a chronic inflammatory disease caused by lipid accumulation, endothelial damage, inflammatory cell infiltration, and plaque formation in the arterial wall [115,116]. In view of the critical role played by ECs in the regulation of the inflammatory response, in blood fluidity, and in vascular tone and permeability, endothelial dysfunction represents an early step in the onset of AS [117,118]. Although the exact cause of AS is unknown, elevated levels of cholesterol and apolipoprotein B, excessive vascular smooth muscle cell (VSMC) proliferation, platelet activation, and inflammatory macrophage recruitment represent major factors impacting on AS progression.

In the context of the vessel wall, exosomes have been shown to modulate crucial processes involved in AS development, mainly related to endothelial functions, proliferation and differentiation of VSMCs, and activation of platelets and macrophages [10]. In addition, it has been shown that exosomes released by macrophages, VSMCs, and platelets, carry miRNAs (such as miR-155 and miR-223) which trigger the activation of the NF-κB inflammatory pathway, enhancing the expression of cell-surface VCAM-1, ICAM-1, and endothelial-leukocyte adhesion molecules; in fact, the consequent EC activation and local inflammation results in exacerbation of AS progression [119,120] (see Table 3). Exosomes can also cause AS progression by fostering immune cell infiltration across the endothelial lining [121,122,123]. In addition, while, at an early stage of AS, cholesterol-enriched exosomes released by macrophages function as protective “scavengers” that allow cholesterol dumping [121], instead, T cell-secreted exosomes promote atherogenesis by increasing cholesterol accumulation in monocytes, as well as by eliciting the release of TNF-alpha and other proinflammatory cytokines controlling vascular cells [124,125,126]. Furthermore, the amount of miR-30e and miR-92a in circulating exosomes is upregulated in AS patients, and negatively correlated with plasmatic cholesterol levels. At the molecular level, these miRNAs have been suggested to act by suppressing the expression of the transporter ATP-binding cassette A1 (ABCA1), a major regulator of cellular cholesterol and phospholipid homeostasis [127], which suggests an interesting role for miR-30e/miR-92a as potential biomarkers for clinical diagnosis and possible targets for the treatment of coronary AS. Interestingly, it has been shown that oxidized LDL, a potent pro-atherosclerotic factor, induces the upregulation of miR-155 expression in VSMCs and its transfer through exosomes to neighboring ECs, hindering their proliferation and migration. Moreover, miR-155 uptake by ECs disrupts vascular endothelial barrier function by suppressing tight junction proteins, thereby facilitating macrophages infiltration and AS [128]. In addition, miR-155 appears to contribute to AS development due to its ability to target the expression of endothelial nitric oxide synthase, which results in altered VSMC activity [129]. In the development of AS, it has also been shown that the activation of CD137 signaling in ECs led to a decreased expression of exosomal TET2, a DNA methylase regarded as regulator of VSMC phenotype. This mechanism enhances the proliferative and migratory phenotype of VSMCs, thus, promoting plaque formation [130].

Inflammatory cells have a pivotal role in AS progression, which also depends on the release of exosome-derived signals in the microenvironment. For instance, metastasis associated in lung adenocarcinoma transcript (MALTA)-11, carried by EC-derived exosomes, promotes the formation of neutrophil extracellular traps (NETs) and M2 macrophage polarization, known to promote AS [131,132]. In turn, exosomal miR-21-3p derived from macrophages inhibits phosphatase and tensin homolog (PTEN) expression and further promotes VSMC migration/proliferation, enhancing AS development [119]. In contrast, several circulating exosome-borne miRNAs, such as miR-126 and miR-199a, appear to be protective against AS [133]. For instance, in a study of 176 patients with stable CAD, elevated levels of miR-126 and miR-199a carried by circulating MVs resulted in a reduced risk of developing unfavorable cardiovascular events, indicating the prognostic relevance of these non-coding RNAs in AS. Moreover, numerous miRNAs carried by EC exosomes have been suggested to reduce AS plaque formation, protecting vascular endothelium from VSMC-derived pathological signals. Studies by Ong [134] and Zernecke [135] demonstrated that hypoxia induced ECs to release exosomes enriched in miRNA-126, miRNA-210, and miR-216, which resulted in inhibition of macrophage infiltration and AS progression. In particular, miR-126 appears to contribute to the stabilization of hardened plaques by unleashing the CXCR4/CXCL12 signaling cascade [135]. Moreover, Hergenreider and colleagues found that the transcription factor KLF2, which is known to mediate an atheroprotective endothelial phenotype induced by the shear stress, regulated the expression of several miRNAs and led ECs to release exosomes enriched in miRNA-143/145. Then, these vesicles were transferred to SMCs, in which their RNA-interfering activity suppressed proliferation and migration. In vivo experiments in ApoE-/- AS mice fed on a high-fat diet further revealed a reduction of atherosclerotic lesions in an miR-143/145-dependent manner [136].

It has been reported that exosomes derived from mesenchymal stem cells may be protective with respect to AS development by inhibiting the expression of miR-342-5p, and upregulating protein phosphatase 1 regulatory subunit 12B (PPP1R12B) [137]. Interestingly, platelet-derived exosomes are enriched in miR-223, miR-339, and miR-21, which have been implicated in the regulation of vascular functions both in vitro and in vivo. In particular, exosomal miR-223 has been shown to inhibit TNF-induced ICAM-1 expression in HUVEC via regulation of the MAPK and NF-κB pathways, indicating its potential role in the regulation of endothelial inflammation and AS development [138].

**Table 3 cancers-14-01969-t003:** Exosomal cargoes involved in vascular-protective and atherosclerotic mechanisms.

Cellular Origin of Exosomes	Cargo	Functions	REFs
Macrophages Vascular smooth muscle cells Platelets	miR-155 miR-223	NF-κB pathway activation ↑ VCAM-1, ICAM-1	[119,121,138]
Vascular smooth muscle cells	miR-155	↓ EC proliferation and migration Tight junction proteins suppression	[128]
Endothelial cells	TET2 MALTA11 miRNA-126 miRNA-210 miR-216 miRNA-143/145	↑ VSMCs proliferation and migration ↑ Plaque formation NETs formation M2 macrophage polarization ↓ Macrophage infiltration ↓ AS progression ↓ SMCs proliferation and migration ↓ Atherosclerotic lesions	[130,131,132,134,135,136]
Macrophages	miR-21-3p	Inhibits PTEN expression ↑ VSMC proliferation and migration ↑ AS development	[119]

Symbols Legend: ↑ increase; ↓ decrease.

It is well accepted that certain pathological conditions, such as diabetes, may increase the risk of developing AS, which represents the main cause of death in diabetic patients [139]. This is due to an imbalance between vasoconstriction and vasodilation [140], and to increased levels of pro-atherogenic reactive oxygen species generated by ECs [141]. Notably, exosomes circulating in the serum of diabetic patients (and db/db mice) contain elevated levels of arginase 1, reducing NO bioavailability in ECs, which suggests a potential role of exosomes in this endothelial dysfunction [142].

Despite the variety of medications available to treat AS, a definite cure for this condition is not available. Notably, the key roles played by exosomes in cell–cell communication and their ability to shuttle molecular cargoes may be of interest to develop alternative approaches for drug delivery [119,143]. Moreover, exosomes released by endothelial progenitor cells (EPCs) have been shown to exert cell protective mechanisms [144,145], and applications of EPC-derived exosomes have been proposed for the treatment of vascular diseases [146]. For example, in an animal model of atherosclerotic diabetes, Bai and collaborators reported that EPC-derived exosomes significantly reduced AS plaques, with a concurrent decrease in inflammatory factors ICAM-1, interleukin-8 (IL-8), C-reactive protein (CRP), as well as oxidative stress factors such as malondialdehyde (MDA) and superoxide dismutase (SOD), which resulted in the amelioration of vascular function [147]. Although such findings support the idea of exosome-based treatment of advanced AS, further studies are required to consolidate this therapeutic perspective.

## 6. Conclusions

The discovery that distant cells can communicate by means of soluble extracellular vesicles, carrying bioactive molecules of different chemical nature, represents a novel standpoint in the study of tissue and organ functions, both in health and in human disease. Accumulating evidence indicates that EVs mediate endothelial cell communication in the microenvironment, although the molecular mechanisms underlying this function in vascular homeostasis and in human disease await elucidation. For example, additional studies are needed to characterize the biological role of exosomes in the regulation of the EC metabolic switch during tumor angiogenesis, and to determine their clinical relevance as pro- or anti-angiogenic mediators. Notably, EVs of different cellular origin are considered to be a valuable source of biomarkers, potentially relevant in the management of diverse pathological conditions, including cancer and atherosclerosis. In addition, a better understanding of the signaling mechanisms mediated by their cargoes may lead to the identification of novel therapeutic targets for the treatment of vascular dysfunction.

## Figures and Tables

**Figure 1 cancers-14-01969-f001:**
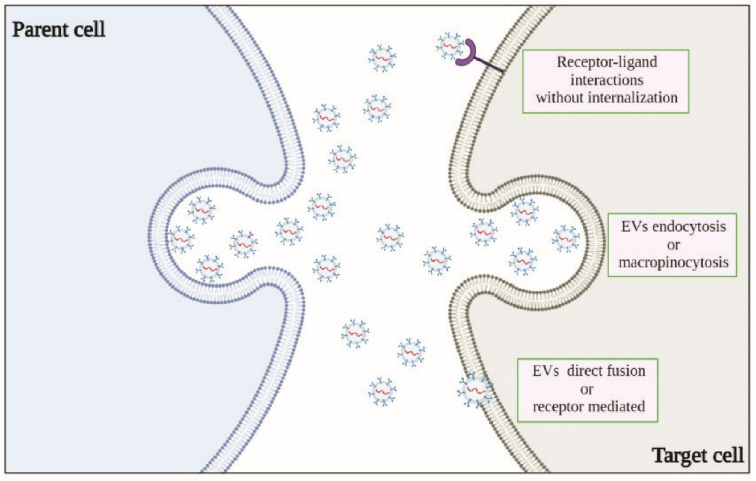
Postulated mechanisms of exosome cell–cell communication.

**Figure 2 cancers-14-01969-f002:**
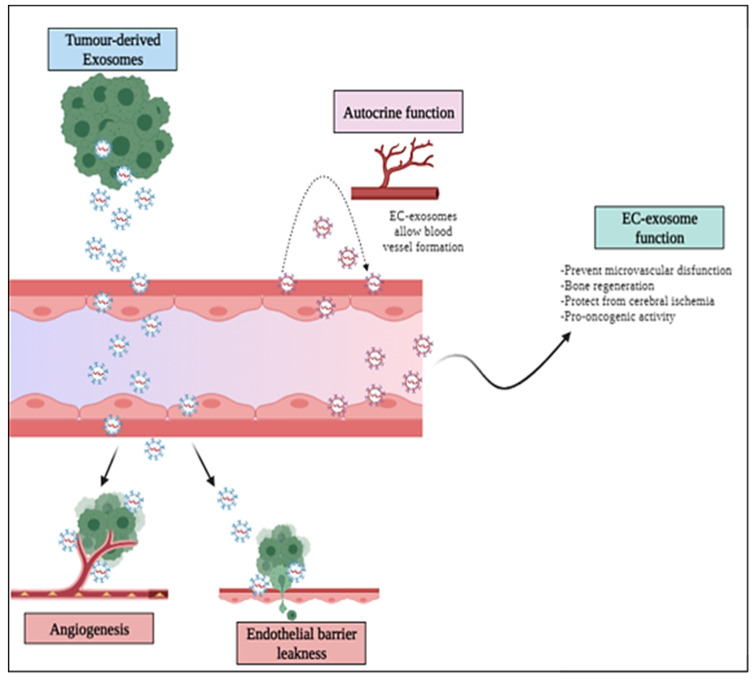
Exosome-mediated regulation of the vasculature in cancer and in other pathologies. Tumor-derived exosomes may induce EC proliferation and migration, promoting the angiogenic process and have been shown to elicit leakiness of endothelial barriers and vascular permeability, thus, fostering cancer cell ingress into the bloodstream for metastatic dissemination. Furthermore, exosomes are released by ECs to self-regulate the same or neighboring cell population, including in the tumor microenvironment.

**Table 1 cancers-14-01969-t001:** Angiogenic proteins carried by tumor-derived exosomes and their functional roles in cancer progression.

Cellular Origin of Exosomes	Angiogenic Proteins	Functional Role	REFs
Glioblastoma	↑ VEGF, TGFβ, βFGF ↑ IL-6, IL-8 ↑ TIMP-1, TIMP-2 ↑ Sema3A	↑ Angiogenesis ↑ Malignancy	[66,71]
Nasopharyngeal carcinoma	↑ MMP-13 ↑ MMP-2 ↑ MMP-9	Metastasis promotion ↑ Proliferation and tube formation in vitro	[73]
Head and neck squamous cell carcinoma	uPA coagulation factorIII MMP-9	↑ Formation of vascular structures ↑ Proliferation of the cells	[75]
Colorectal carcinoma	Tetraspanin-8 Plexin B2	↑ Angiogenesis	[76]
Lung adenocarcinoma	Sortilin	↑ Angiogenic protein	[77]
Melanoma	VE-Cadherin uPAR EGFR	↑ Angiogenesis in vitro and in vivo	[78,79]
Breast cancer	Annexin II	Metastasis and Angiogenesis promotion	[80]
Myeloma, lymphoblastoid and breast cancer	Heparanase	↑ Angiogenic factors and tube formation	[81]

Symbols Legend: ↑ increase.

## Data Availability

Data sharing not applicable. No new data were created or analyzed in this study. Data sharing is not applicable to this article.

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
