# Peer review of "Message in a Bottle: Endothelial Cell Regulation by Extracellular Vesicles"

_cancers, 2022, doi:10.3390/cancers14081969_

Round 1

Reviewer 1 Report

The manuscript by Palazzo et al provides an overview about the role of extracellular vesicles (exosomes, microvesicles) in cell-cell communication at the endothelial level. The manuscript is clear and well written. The following aspects should be improved before publication:

1-the whole word "exosome" or "exosomes" should be used instead of the abbreviation exo.

2- Table 2, the title has to be reformulated to include other RNA species, which are not miRNAs. I suggest replacing miRNAs by small RNAs.

3- Table 2, first line of the table. References 82-86 do not cover only colorectal cancer but also other malignancies (e.g. lung cancer). This should also be mentioned in the table. 

4- The abbreviation circ-IARS needs to be defined.

5- Tables describing the donor cells, cargo and effect on the target cells (similar to tables 1 and 2) should be added for section 5.

Author Response

The manuscript by Palazzo et al provides an overview about the role of extracellular vesicles (exosomes, microvesicles) in cell-cell communication at the endothelial level. The manuscript is clear and well written. The following aspects should be improved before publication:

1-the whole word "exosome" or "exosomes" should be used instead of the abbreviation exo.

Author Response: We thank the Reviewer for this advice, and made this change throughout the revised manuscript.

2- Table 2, the title has to be reformulated to include other RNA species, which are not miRNAs. I suggest replacing miRNAs by small RNAs.

AR: We thank the Reviewer for this advice. We changed the title of the table in the revised manuscript.

3- Table 2, first line of the table. References 82-86 do not cover only colorectal cancer but also other malignancies (e.g. lung cancer). This should also be mentioned in the table.

AR: We thank the Reviewer for rightfully raising this point. We modified the table accordingly.

4- The abbreviation circ-IARS needs to be defined.

AR: We reckon that it is appropriate to define all abbreviations in the text. However, as far as we can understand from scientific literature, “circ-IARS” actually represents the proper name (or “gene symbol”) attributed to this small RNA, and not an acronym of specific significance.

5- Tables describing the donor cells, cargo and effect on the target cells (similar to tables 1 and 2) should be added for section 5.

AR: We thank the Reviewer for this advice. We have generated a new Table 3, summarizing the indicated content of section 5.

Reviewer 2 Report

This is a well written and informative paper. I only have a few suggestions.

  1. The paper has a lot of unnecessary abbreviations (like ILV, Exo, GSC, TEX, etc.) that should be eliminated.
  2. While the paper is very well written, it needs better proof reading and English usage editing. As examples,

Line 68, change adopted to adapted.

Line 75, size should be sizes

Line 309, involved should be implicated

  1. The legend for Fig. 2 should be expanded so that it is self-explanatory
  2. The authors should add to their discussion of the data compiled in Table 2. Why isn’t the same miRNA identified in multiple different cancers? What is the strength of evidence supporting each? Is some more compelling than others?
  3. Section 5 on exosomes in other cases besides cancer needs some more organization or should be deleted. The relationship of this section to the rest of the paper is unclear to me. I do not see any conclusions or inferences from it that help me. It more of a catalog than a critique.
  4. The sentence (line 502 and ff) should be deleted, unless the authors add more about EVs and drug delivery.

Author Response

This is a well written and informative paper. I only have a few suggestions.

The paper has a lot of unnecessary abbreviations (like ILV, Exo, GSC, TEX, etc.) that should be eliminated.

Author Response: We thank the Reviewer for this advice. We removed all unnecessary abbreviations throughout the revised manuscript.

While the paper is very well written, it needs better proof reading and English usage editing. As examples,

Line 68, change adopted to adapted.

Line 75, size should be sizes

Line 309, involved should be implicated

AR: We thank the Reviewer for noticing these mistakes that we have now corrected in the revised manuscript. However, we did not change the word “adopted” at line 68, since this is indeed the intended term, meaning “assumed”, “accepted” (to define something).

The legend for Fig. 2 should be expanded so that it is self-explanatory

AR: We thank the Reviewer for this advice. We have expanded the figure legend, as requested.

The authors should add to their discussion of the data compiled in Table 2. Why isn’t the same miRNA identified in multiple different cancers? What is the strength of evidence supporting each? Is some more compelling than others?

AR: We thank the Reviewer for appropriately raising this issue. Actually, although the study of exosome-mediated communication within the tumor microenvironment is particularly exciting, our knowledge on EC-regulating small RNAs carried by tumor-derived exosomes needs further development. It might be postulated that the wide spectrum of functionally active small RNA species found in the exosomes released by different tumor cells reflects tissue-specificity. However, in page 9, we have added a statement discussing this issue, and underscoring the need for further researches addressing the consistency across human cancers of the reported relevant candidates, which could lead to the identification of common biomarkers and most significant exosome-borne miRNAs regulating tumor vasculature.    

Section 5 on exosomes in other cases besides cancer needs some more organization or should be deleted. The relationship of this section to the rest of the paper is unclear to me. I do not see any conclusions or inferences from it that help me. It more of a catalog than a critique.

AR: We thank the Reviewer for her/his advice on improving this part of the manuscript. We decided to focus Section 5 on the role of exosomes in the pathogenesis of vascular atherosclerosis, following a structure and rationale similar to that applied in the part concerning tumor vasculature, and including a novel Table 3. The topic was also mentioned in the revised Abstract. As in case of cancer patients, the clinical relevance of the findings in this emerging research field is both for the identification of targetable mechanisms, as well as in the perspective to identify novel circulating biomarkers valuable for disease staging and monitoring.

The sentence (line 502 and ff) should be deleted, unless the authors add more about EVs and drug delivery.

AR: We thank the Reviewer for this advice, and have revised the conclusions and deleted the indicated statement.

Reviewer 3 Report

The authors provided a well-documented overview of Ev's role in the communication mechanisms between cancer and endothelial cell within the tumor microenvironment. In addition to this, the review could represent a really interesting point of view in a field so dynamic and rich in potential future applications. The field of research focused on exosomes is in continuous evolution and even if the article is well written, the introduction section could be improved by adding some recent works related to the need for new technologies able to associate a specific marker with an exosome subtype and this exosome subtype to a particular function and/or group of functions (PMID: 35141731 is just an example). 

I hope that my comments could be useful and I look forward to reading the revised version of the paper.

Good luck.

Author Response

The authors provided a well-documented overview of Ev's role in the communication mechanisms between cancer and endothelial cell within the tumor microenvironment. In addition to this, the review could represent a really interesting point of view in a field so dynamic and rich in potential future applications. The field of research focused on exosomes is in continuous evolution and even if the article is well written, the introduction section could be improved by adding some recent works related to the need for new technologies able to associate a specific marker with an exosome subtype and this exosome subtype to a particular function and/or group of functions (PMID: 35141731 is just an example).

Author Response: We thank the Reviewer for her/his appreciation of our manuscript. Surely a number of issues remain open in the field, especially in perspective to application in translational medicine. The question about heterogeneity of circulating exosomes found in patient liquid biopsies is particularly challenging. In fact, this is potentially advantageous, inasmuch it can provide a better picture of intratumor heterogeneity at systemic level; however, in other respects, it may be needed a specific and accurate sorting of cancer cell-derived exosomes from those shed by non-tumor cells. In this review manuscript, we mainly focused on signaling aspects of exosomes controlling endothelial cell behavior, especially in vascular dysfunction. However, following Reviewer’s suggestion, we have mentioned the importance of this open issue at the end of introductory Section 2, also alluding to the need for developing appropriate technological approaches for exosome subtype discrimination.